# Concurrence: A dependence criterion for time series, applied to biological data

## Abstract

Measuring the statistical dependence between observed signals is a primary tool for scientific discovery. However, biological systems often exhibit complex non-linear interactions that currently cannot be captured without a priori knowledge or large datasets. We introduce a criterion for dependence, whereby two time series are deemed dependent if one can construct a classifier that distinguishes between temporally aligned vs. misaligned segments extracted from them. We show that this criterion, concurrence, is theoretically linked with dependence, and can become a standard approach for scientific analyses across disciplines, as it can expose relationships across a wide spectrum of signals (fMRI, physiological and behavioral data) without ad-hoc parameter tuning or large amounts of data.

## 1 Introduction

Measuring the statistical dependencies between biological signals is fundamental for understanding the complex interplay within and between molecular, neurobiological, and behavioral processes. The most common approach to quantifying dependence is using linear model-based statistics, with the Pearson correlation coefficient being the dominant metric (Tjøstheim et al., 2022). However, biological systems often exhibit interactions (Janson, 2012) that cannot be captured by linear models (He & Yang, 2021), such as cross-frequency coupling (Schmidt & O'Brien, 1997), threshold effects (Beltrami & Jesty, 1995), phase shifts (Tiesinga & Sejnowski, 2010), feedback systems (Beltrami & Jesty, 1995), or multi-scale interactions (Qu et al., 2011).

While linear models cannot comprehensively capture statistical dependence, linear and non-linear models together can. That is, if two time series $x$ and $y$ are dependent but uncorrelated, then there must be (non-linear) mathematical transformations $f$ and $g$ such that the transformed signals $f(x)$ and $g(y)$ are correlated (Rényi, 1959). However, the transformations that expose the dependence can be particular to each problem and difficult to identify when the compared signals are generated by complex or unknown mechanisms. The Hilbert-Schmidt Independence Criterion (Gretton et al., 2007) –which can be considered as a generalization of distance correlation (Székely et al., 2007; Sejdinovic et al., 2013)– or variants of canonical correlation analysis (Verbeek et al., 2003; Andrew et al., 2013) can, in principle, determine linear and non-linear dependence. However, these approaches are successful only if one can identify model parameters or kernels that expose the dependence (Hua & Ghosh, 2015; Gretton et al., 2012), which may not be possible or may require large samples (Zhuang et al., 2020; Marek et al., 2022). Alternatively, one may use analytical transformations such as Fourier or wavelet decomposition (Greenblatt et al., 2012; Fujiwara & Daibo, 2016; Schmidt et al., 2012), but the generalizability of this approach is limited, as there is no single transformation that works for all signals (Mallat, 2009; Vetterli et al., 2014). Moreover, analytical transformations pose family-wise error problems (Maraun & Kurths, 2004; Kramer et al., 2008) because they typically decompose each signal into multiple signals (e.g., frequency bands), and dependence can occur between any pair of decomposed signals (e.g., cross-frequency dependence). These issues are exacerbated when the compared signals are multi-dimensional and only a subset in one set of signals depends on an unknown subset in the other. In sum, currently there is no tractable method that can detect or quantify the dependence between a broad variety of biological signals when the dependence structure is not known a priori—presenting a major obstacle to scientific discovery.

We introduce a new approach, called *concurrence*, to quantify the statistical dependence between pairs of signals. The proposed approach is based on a simple idea: if two signals are statistically

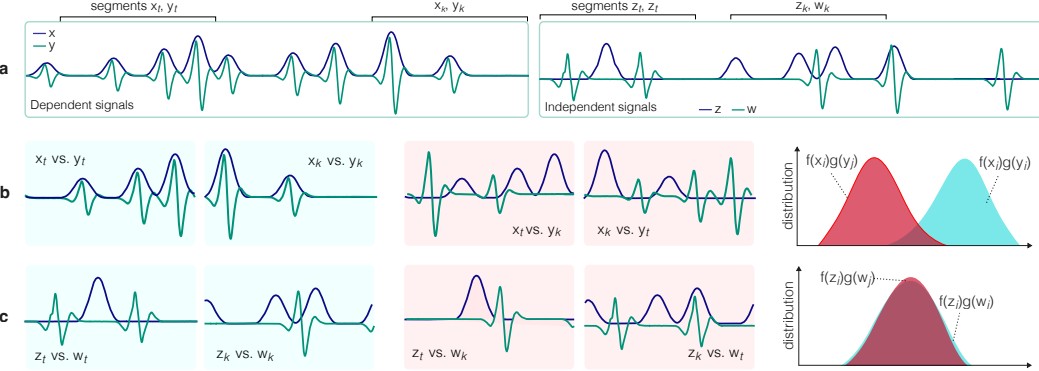

Figure 1: (a) Dependent signals $x$ and $y$, and independent signals $w$ and $z$. (b) Concurrent segments from $x$ and $y$ have different characteristics from non-concurrent segments, thus one can find functions $f$ and $g$ such that $f(x_i)g(y_j)$ is, on average, larger for concurrent segments (i.e., $i = j$) compared to non-concurrent segments (e.g., $f$ as the identity operator and $g$ the integral operator). (c) Concurrent and non-concurrent segments extracted from independent signals are not statistically distinguishable.

dependent, then temporally aligned (i.e., concurrent) segments of the compared signals are expected be separable from segments that are temporally misaligned (i.e., not concurrent), as illustrated in Fig. 1. We show that this idea is not only intuitive, but also theoretically plausible, and it provides a straightforward and powerful recipe for automatically finding linear or non-linear transformations that expose dependence—namely, training a machine learning model that classifies between concurrent vs. non-concurrent segments from signals. Concurrence is essentially a contrastive learning (CL) approach that is not only used for representation learning, but also defines a bounded coefficient for statistical dependence (Section 2.3). That is, the proposed approach yields a score coined the concurrence coefficient, which is scaled between 0 and 1. Theoretical and experimental results indicate that this coefficient is proportional to the degree of dependence between the compared signals.

We apply concurrence to three distinct types of biological signals (i.e.,, fMRI, physiological, and behavioral data). Results suggest that concurrence can become a standard tool for scientific analyses, as it satisfies two critical priorities. First, the dependence in all compared biological signals as well as a large number of challenging synthetic datasets is detected without *any* ad-hoc hyperparameter modification, thus analyses can be conducted without the loss of power that would be caused by correcting for each tested parameter. Second, dependence can be detected even from modestly sized datasets without any pre-training, suggesting that concurrence can be applied in scientific domains where data is scarce or hard to obtain, or involve arbitrary types of sensors. We created an easy-to-use open-source software that implements concurrence, and runs efficiently on a single GPU. We will make this implementation publicly available should this paper be accepted for publication.

## 2 CONCURRENCE

Suppose that $x_{t,w}$ and $y_{t,w}$ are segments of signals $x$ and $y$, observed between the time points $t$ and $t + w$. If both $x_{t,w}$ and $y_{t,w}$ contain responses to a common event, then they must be statistically dependent. Thus, there must exist transformations $f$ and $g$ such that transformed representations of the segments, $f(x_{t,w})$ and $g(y_{t,w})$, are correlated (Rényi, 1959). The crux of our approach is that, while $f(x_{t,w})$ and $g(y_{t,w})$ are expected to be correlated, $f(x_{t,w})$ and $g(y_{t',w})$ are, on average, uncorrelated if $t'$ is a random time point, different from $t$ (Fig. 1b).

This idea is not only intuitive, but also theoretically plausible (Section 2.2), and provides a straightforward manner of detecting dependence automatically via self-supervision. Specifically, we quantify dependence via the *concurrence coefficient*, which is obtained by training a classifier (Section 2.1) to distinguish between concurrent vs. non-concurrent segments cropped randomly from a dataset of signal pairs, and calculating the normalized classification accuracy on another dataset $\mathcal{D} = \{(x^i, y^i)\}_i$ :

$$\text{concurrence coefficient} = 2 \times \max(\text{accuracy}, 0.5) - 1. \tag{1}$$

The concurrence coefficient is bounded between 0 and 1, and its magnitude is proportional to the degree of dependence between the compared signal pairs (Section 2.2 and Fig. 2). The dataset $\mathcal{D}$ must not overlap with the dataset used during training, lest the classifier may overfit and overestimate the dependence. Thus, one may use cross-validation (CV) and compute the average concurrence coefficient over the test sets of CV folds, or compute the concurrence coefficient on independent data.

## 2.1 THE LOSS FUNCTION AND THE PER-SEGMENT CONCURRENCE SCORE

The classifier that we use to compute the concurrence coefficient is a neural network that produces a *per-segment concurrence score* (PSCS) from segments $x_{t,w} \in \mathbb{R}^{K_x \times w}$ and $y_{t',w} \in \mathbb{R}^{K_y \times w}$, where $K_x$ and $K_y$ are the dimensions of $x$ and $y$. The two segments are deemed concurrent if the PSCS is positive, and not concurrent if it is nonpositive. The PSCS is computed by first transforming the input segments via separate learned functions $f$ and $g$ into segments of dimensions $K_f$ and $K_g$,

$$f : \mathbb{R}^{K_x \times w} \to \mathbb{R}^{K_f \times w'}, \ g : \mathbb{R}^{K_y \times w} \to \mathbb{R}^{K_g \times w'} \tag{2}$$

where $w'$ is the temporal length of the transformed segments; then computing the covariance of the transformed segments, $C = \mathrm{Cov}\left(f(x_{t,w}), g(y_{t',w})\right)$; and finally calculating the weighted average of the entries of the covariance matrix

$$s = \sum_i \sum_j \alpha_{ij} C_{ij}, \tag{3}$$

where $C_{ij}$ is the $ij$th entry of $C$. The weights $\alpha_{ij}$, as well as the transformations $f$ and $g$, are learned from scratch while training the network. The loss function is simply the binary cross entropy (with logits), which is used to classify between PSCS values computed from randomly extracted concurrent segments ($t = t'$) versus non-concurrent segments ($t \neq t'$).

In addition to computing the concurrence coefficient, the PSCS can also be used one to quantify the dependence between a specific pair of segments. As such, the concurrence coefficient and the PSCS have two distinct uses for scientific analyses. While the concurrence coefficient can uncover whether and to what extent two processes (e.g., breathing rate and cardiac activity) are related in general (i.e., at the sample level), the PSCS between specific pairs of concurrent segments (i.e., $t = t'$) can uncover whether this relationship is stronger for a specific individual, for individuals with a certain condition (e.g., anxiety), or for certain moments within the compared signals. Our experiments on real data include use cases for both the concurrence coefficient and the PSCS.

## 2.2 THEORETICAL RELATION BETWEEN DEPENDENCE AND CONCURRENCE

We theoretically show that if the pairs of signals in $\mathcal{D}$ are statistically dependent, then concurrent segments extracted from them are expected to be separable from non-concurrent segments. Moreover, the degree of separability increases with the degree of dependence between the signals, and with the segment size $w$. For tractability, our theoretical analysis relies on some simplifying assumptions— that the signals rely on stationary (Bernoulli) processes, and that there is no time lag between the dependent signals. Nevertheless, even these assumptions lead to challenging cases of non-linear dependence (Appendix E), and numerical simulations (Appendix C) and experiments with synthetic data show that the results of the theoretical analysis hold even when these assumptions are violated.

Suppose that the signals $x^i$ and $y^i$ in $\mathcal{D}$ are realizations of respective discrete-time random sequences (RSs) $\mathbf{x}[t]$ and $\mathbf{y}[t]$ that are generated by convolving two binary RSs $\mathbf{h}_x[t]$ and $\mathbf{h}_y[t]$ with kernels $k_1[t]$ and $k_2[t]$, and adding noise processes $\mathbf{n}_x[t]$ and $\mathbf{n}_y[t]$ (Supp. Fig. E.1):

$$\mathbf{x}[t] = (\mathbf{h}_x \star k_1)[t] + \mathbf{n}_x[t] \tag{4}$$

$$\mathbf{y}[t] = (\mathbf{h}_y \star k_2)[t] + \mathbf{n}_y[t]. \tag{5}$$

The noise processes $\mathbf{n}_x[t]$ and $\mathbf{n}_y[t]$ are assumed to be independent from each other and from $\mathbf{h}_x[t]$ or $\mathbf{h}_y[t]$. The processes $\mathbf{h}_x[t]$ and $\mathbf{h}_y[t]$ are modeled as the product of a common process $\mathbf{h}[t]$ with two separate and independent processes $\boldsymbol{\alpha}[t]$ and $\boldsymbol{\beta}[t]$ that take binary values (0 or 1):

$$\mathbf{h}_x[t] = \boldsymbol{\alpha}[t]\mathbf{h}[t] \tag{6}$$

$$\mathbf{h}_y[t] = \boldsymbol{\beta}[t]\mathbf{h}[t], \tag{7}$$

where $\mathbf{h}[t]$, $\boldsymbol{\alpha}[t]$ and $\boldsymbol{\beta}[t]$ are Bernoulli processes with respective parameters $p$, $p_\alpha$, $p_\beta$. The common process $\mathbf{h}[t]$ ensures that $\mathbf{x}[t]$ and $\mathbf{y}[t]$ are dependent, provided that $p \neq 0$, $p_\alpha \neq 0$ and $p_\beta \neq 0$. The processes $\boldsymbol{\alpha}[t]$ are $\boldsymbol{\beta}[t]$ make the dependence stochastic when $p_\alpha, p_\beta \in (0,1)$, as the latter implies that only an a priori unknown set of events (i.e., $\mathbf{h}[t] = 1$) will be observed both in $\mathbf{x}[t]$ and in $\mathbf{y}[t]$.

Simple metrics such as correlation may be unable to expose even a deterministic dependence between $\mathbf{x}$ and $\mathbf{y}$, as the kernels $k_1$ and $k_2$ can make the dependence non-linear. If one could recover the underlying binary processes $\mathbf{h}_x$ and $\mathbf{h}_y$, one could simply compute the inner product of these two processes to expose dependence. While perfect recovery may not be realistic (e.g., due to noise $\mathbf{n}_x$ and $\mathbf{n}_y$), one can assume that there exist estimators $\tilde{\mathbf{h}}_x$ and $\tilde{\mathbf{h}}_y$ that estimate these binary events (e.g., through deconvolution) up to additive error terms $\boldsymbol{\epsilon}_x[t]$ and $\boldsymbol{\epsilon}_y[t]$ as

$$\tilde{\mathbf{h}}_x[t] = \min\{\boldsymbol{\alpha}[t]\mathbf{h}[t] + \boldsymbol{\epsilon}_x[t], 1\} \tag{8}$$

$$\tilde{\mathbf{h}}_y[t] = \min\{\boldsymbol{\beta}[t]\mathbf{h}[t] + \boldsymbol{\epsilon}_y[t], 1\}, \tag{9}$$

where $\boldsymbol{\epsilon}_x[t]$ and $\boldsymbol{\epsilon}_y[t]$ are Bernoulli processes with respective parameters $p_x^\epsilon$ and $p_y^\epsilon$, and the $\min\{\cdot\}$ operator ensures that $\tilde{\mathbf{h}}_x[t]$ and $\tilde{\mathbf{h}}_y[t]$ is a binary RS. The theorem below expresses that if $\mathbf{x}$ and $\mathbf{y}$ are dependent, then the inner product between $\tilde{\mathbf{h}}_x$ and $\tilde{\mathbf{h}}_y$ is larger when computed when these processes are temporally aligned compared to when they are misaligned.

**Theorem 1.** *Suppose that $\mathbf{z}(\tau)$ is an RV defined as a function of a temporal lag parameter $\tau$ as*

$$\mathbf{z}(\tau) = \frac{1}{w}\sum_{t=1}^{w} \min\{\boldsymbol{\alpha}[t]\mathbf{h}[t] + \boldsymbol{\epsilon}_x[t], 1\} \min\{\boldsymbol{\beta}[t+\tau]\mathbf{h}[t+\tau] + \boldsymbol{\epsilon}_y[t+\tau], 1\}, \tag{10}$$

*where $\mathbf{h}$, $\boldsymbol{\alpha}$, $\boldsymbol{\beta}$, $\boldsymbol{\epsilon}_x$ and $\boldsymbol{\epsilon}_y$ are Bernoulli processes with respective parameters $p$, $p_\alpha$, $p_\beta$, $p_x^\epsilon$ and $p_y^\epsilon$ such that*

$$p \in (0,1), \ \ p_\alpha, p_\beta \in (0,1], \ \ and \ \ p_x^\epsilon, p_y^\epsilon \in [0,1). \tag{11}$$

*Further, suppose that $\mathbf{z}^+$ and $\mathbf{z}^-$ are RVs derived from $\mathbf{z}(\tau)$ by respectively setting $\tau$ to zero and to a fixed and nonzero value $\tau' \neq 0$:*

$$\mathbf{z}^- = \mathbf{z}(\tau'), \ \ \mathbf{z}^+ = \mathbf{z}(0). \tag{12}$$

*Then, $\mathbb{E}\{\mathbf{z}^+\} > \mathbb{E}\{\mathbf{z}^-\}$ for any time window $w \in \mathbb{Z}^+$, and the difference between the two means is*

$$\mathbb{E}\{\mathbf{z}^+\} - \mathbb{E}\{\mathbf{z}^-\} = p(1-p)p_\alpha p_\beta (1 - p_x^\epsilon)(1 - p_y^\epsilon). \tag{13}$$

*Moreover, there exists a threshold $\theta$ such that $P\{\mathbf{z}^+ < \theta\} \to 0$ and $P\{\mathbf{z}^- > \theta\} \to 0$ as $w \to \infty$.*

The proof is provided in Appendix B. Theorem 1 implies that one can predict beyond chance level if segments extracted from the signal pairs in $\mathcal{D}$ are concurrent or not, as the distribution of the $\mathbf{z}^+$ is not fully overlapping with that of $\mathbf{z}^-$ (Supp. Fig. C.2). Also, the concurrence coefficient is expected to be proportional to the degree of dependence between the compared segments, as the difference in equation 13 increases with $p_\alpha$ and $p_\beta$. The latter difference decreases as $p_x^\epsilon$ and $p_y^\epsilon$ increases, thus the concurrence coefficient is expected to be inversely related to the amount of noise. Finally, the concurrence coefficient is expected to also increase with the segment size, as the overlap between the distributions of $\mathbf{z}^+$ and $\mathbf{z}^-$ decreases with increasing $w$. The exact $w$ value is expected to have little significance for exposing dependence, as the distributions of $\mathbf{z}^+$ and $\mathbf{z}^-$ are theoretically separable for any positive $w$. Experiments on synthetic data corroborate this theoretical analysis (Fig. 2).

## 2.3 Comparison with Other Contrastive Learning Approaches

Concurrence relies on CL, and it is by no means the first CL approach on time series (Zhang et al., 2024). However, its goal and operation are fundamentally different from existing approaches, which typically use CL for learning a representation that can be used for system identification (Hyvarinen & Morioka, 2016); or for downstream tasks, such as classification, forecasting, clustering or anomaly detection (Woo et al., 2022; Zhang et al., 2022; Tonekaboni et al., 2021; Oord et al., 2018; Tian et al., 2020). In contrast, the the primary contribution of our work is to show that our CL criterion (i.e., concurrence) provides a principled and naturally bounded metric of statistical dependence for time series—a metric that is suitable for scientific analyses, as it exposes a wide range of dependencies without requiring users with deep learning expertise, large amounts of data or ad-hoc parameter tuning.

## 3  IMPLEMENTATION

**Network architecture.** To make the concurrence approach useful for scientific purposes, the functions $f$ and $g$ should be flexible enough to expose arbitrary dependencies. Also, one needs a training procedure that requires no hyperparameter tuning and can work successfully even with modestly-sized datasets, since the samples used in scientific analyses often have only hundreds or even fewer samples. As such, we model the transformations $f$ and $g$ with Convolutional Neural Networks (CNNs), which are universal approximators (Zhou, 2020) and, thanks to advances in machine learning (Wang et al., 2016; Bai et al., 2018), have well-established recipes for training across a large variety of temporal analysis tasks without ad-hoc modifications (Wang et al., 2016), particularly when modeling short-term dependencies (Bai et al., 2018). Our experiments with real and synthetic data verify that CNNs with the *same parameters* (Appendix A) can detect a wide range of linear or non-linear dependence patterns between signals that have distinct frequency characteristics and are corrupted by large amounts of noise. Further, the training does not require an unrealistic sample size, as our experiments show that even fewer than 100 signal pairs can suffice.

**Significance testing.** We advise using significance tests to avoid false discoveries (i.e., Type I error), particularly when the concurrence coefficient is computed on small samples. Specifically, we recommend permutation tests (Nichols & Holmes, 2002) where the null distribution is constructed from concurrence coefficients computed after randomly re-assigning the labels of segments pairs (i.e., concurrent/non-concurrent). Our analyses (Appendix D) show that such tests can be conducted efficiently by using a relatively small number (e.g., 1000 or fewer) of permutations (Winkler et al., 2016) and leveraging a Pearson Type III approximation (Kazi-Aoual et al., 1995).

**Computational complexity.** The complexity of our implementation is $\mathcal{O}\left(Nw(K_x + K_y)\right)$, where $N$ is the number of signals used for training the classifier. As a reference, computing the concurrence coefficient for each of the synthesized datasets ($T = 1000$, $N = 400$, $w = 400$; see Section 4.2) takes approximately 40 seconds on a single NVIDIA RTX 3090 GPU.

## 4  EXPERIMENTAL VALIDATION

We first investigate whether the theoretically expected properties of concurrence hold in practice. Next, we compare concurrence with eight alternative methods on a large number of synthesized datasets. Finally, we apply concurrence on three types of real data with single- and multi-dimensional signals that are dependent linearly or non-linearly, namely brain imaging (fMRI), physiological (breathing and heart rate), and behavioral (facial expressions and head movements) signals.

### 4.1  MEASURING STOCHASTIC DEPENDENCE, AND THE EFFECT OF SEGMENT SIZE

Biological signals may depend on each other stochastically rather than deterministically, as they typically reflect an admixture of multiple processes, and only a subset of these processes may be dependent between compared signals. Another cause of non-deterministic dependence is measurement noise, which can overpower the compared signals (Welvaert & Rosseel, 2013).

Fig. 2a simulates scenarios where the degree of dependence is controlled with a parameter $\xi$, which varies between 0 (independent signal pairs) and 1 (completely dependent signal pairs). Fig. 2c shows the concurrence coefficients obtained from pairs of signals with varying degrees of dependence (i.e., $\xi$). Results show that the concurrence coefficient successfully detects dependencies ($\xi > 0$) and lack thereof ($\xi = 0$). Of note, the concurrence coefficient is approximately linearly proportional to $\xi$ when $w$ is large enough to contain the entire dependence event (e.g., $w = 100$ for signals in Fig. 2a) but not much larger. While the degree of dependence $\xi$ is overestimated with larger segments, there is no risk of detecting spurious relationships (false positives), as the concurrence coefficient remains approximately zero when $\xi = 0$, regardless of $w$. Fig. 2d shows that concurrence can also handle noise. The concurrence coefficient usually decreases with the signal-to-noise (SNR), yet it can uncover dependence even when the SNR is 0.10, which is lower than a worst-case estimate of noise in fMRI data (SNR=0.35; see Welvaert & Rosseel (2013)).

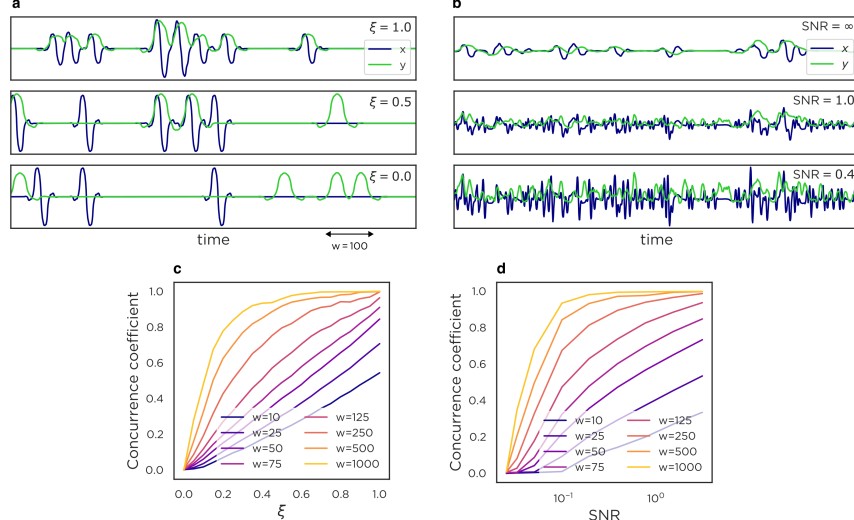

Figure 2: (a) Synthesized signals with deterministic dependence ($\xi = 1.0$), stochastic dependence ($\xi = 0.5$) and no dependence ($\xi = 0.0$). (b) Dependent pairs of signals with varying degrees of noise. (c) Concurrence coefficient vs. $\xi$ (d) Concurrence coefficient vs. signal-to-noise ratio (SNR).

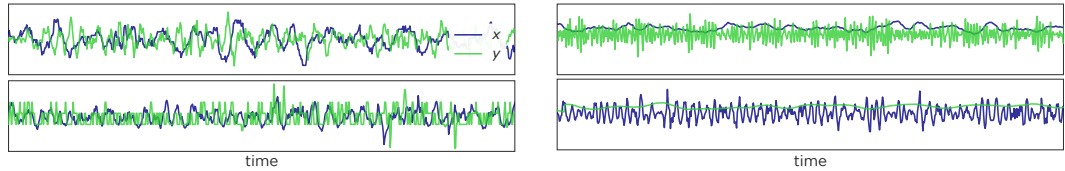

Figure 3: Pairs of (dependent) signals from six of the 100 synthesized datasets used in experiments. Dependence is typically not easy to visually ascertain.

## 4.2 COMPARISON WITH ALTERNATIVE METHODS

We generated 100 synthetic datasets, where each dataset contained pairs of statistically dependent signals. The goal was to determine the dependence in as many datasets as possible by using only the off-the-shelf implementation of our algorithm, without any ad-hoc parameter adjustment. The datasets were designed to be challenging, with dependencies difficult to visually ascertain (Fig. 3).

**Datasets.** Each of the 100 synthesized datasets is comprised of 500 pairs of signals $(x, y)$ generated by convolving a randomly generated binary signal with a randomly determined kernel, akin to equations 4 and 5. Specifically, the kernels were randomly chosen wavelets (at random scales) from the `pywavelets` library, typically resulting in non-linear dependence. To test the algorithm in the presence of non-stationarity, the produced binary signals were made non-stationary, with the probability of observing an event increased or decreased linearly over time at a random rate. To simulate lagged dependence, one of the signals in each pair was also shifted (circularly) by a random lag between 0 and 50 time frames. We also added noise to each signal, by generating it in a similar way—namely, by convolving randomly selected kernels with separate and independent binary signals.

**Compared methods**. We compared concurrence with correlation (Pearson's $r$), windowed cross-correlation (WCC) (Boker et al., 2002), distance correlation (DC) (Székely et al., 2007), Hilbert-Schmidt Independence Criterion (HSIC) (Gretton et al., 2007), Mutual Information (MI), Conditional MI (CMI), Multiscale Graph Correlation (MGC), Kernel Mean Embedding Random Forest (KMERF). HSIC, MGC and KMERF have been implemented via the `hyppo` software package; DC was implemented via `dcor` (Ramos-Carreño, 2022); and Pearson's $r$ was implemented through `scikit-learn`. We provided our own implementation for the remaining methods. The statistical significance for all methods have been computed via permutation tests. Concurrence coefficient was computed on 20% of each dataset, after using 80% of data for training.

Table 1: Results from the 100 synthetic datasets: The number of datasets that have been (correctly) identified as statistically dependent (at significance level 0.05) by each of the compared methods.

| Pearson's $r$ | WCC | DC | HSIC | MI | CMI | MGC | KMERF | Concurrence |
|---|---|---|---|---|---|---|---|---|
| 8 | 10 | 12 | 10 | 7 | 34 | 9 | 11 | 97 |

**Results**. Table 1 shows the results of experiments on synthesized data. CMI is the best among methods alternative to concurrence, due possibly to its ability to model non-linear dependence and temporal dependence, yet can detect the dependence in only 34% of the datasets. All alternative methods can possibly detect the dependence in more datasets if their parameters are optimized for each dataset. However, this is often not feasible for scientific analyses with modestly sized datasets, as one should do multiple tests correction (Armstrong, 2014) for the tested parameter values, leading to significant decrease in statistical power. Concurrence detected the dependence in 97% of datasets, using identical hyperparameters (Appendix A) and segment size ($w = 400$).

## 4.3 Applications to Real Biological Signals

**Brain Imaging** Our experiments on fMRI signals aim to identify how strongly different brain regions are functionally connected. Pearson's $r$ is the single-most commonly used metric for this purpose (Liu et al., 2024). Fig. 4 compares the connectivity matrices obtained with Pearson's $r$ (i.e., correlation matrix) and the concurrence coefficient (i.e., concurrence matrix) on a version of the Philadelphia Neurodevelopmental Cohort dataset (Baum et al., 2020) that was pre-processed as in prior work (Satterthwaite et al., 2016). This dataset uses the parcellation scheme that divides each brain into 400 regions (Schaefer et al., 2018). The concurrence coefficient is computed on segments of size $w = 30$ time points, which corresponds to approximately 90 seconds, whereas the entire signals included 120 time points. Thirty percent of the dataset (426 participants) was used to train the neural networks needed for the concurrence coefficients, and the results in both connectivity matrices were computed from the remaining 70%. The overall similarity between the two connectivity matrices (Fig. 4c vs. Fig. 4d) is striking and suggests that the concurrence coefficient uncovers a dependence structure that has been validated in the field. Fig. 4g shows that there are no pairs of regions with a concurrence score less than 0.2, even though there are many pairs that are uncorrelated (i.e., Pearson r $\approx$ 0), suggesting that concurrence captured statistical dependencies that cannot be captured with correlation (e.g., Fig. 4b) as well as those that can (e.g., Fig. 4a). The fact that the concurrence coefficient exposed a dependence between all 400×199=79,600 pairs of brain regions with 79,600 independently trained networks verifies that the training needed for concurrence can be done robustly. We ran a permutation test to identify if the method detects spurious dependence (Type I error) by computing concurrence between signals of mismatched participants. The concurrence coefficient was closely distributed around zero (Fig. 4h), indicating no spurious relationships. The differences between the concurrence coefficient and Pearson's correlation exhibit a structured pattern across the seven brain networks (Fig. 4F), increasing progressively from lower-order affective (limbic), somatomotor and sensory (visual) networks to higher-order cognitive control (ventral attention, dorsal attention, default mode, frontoparietal) networks. This systematic increase suggests that linear correlation may not be capturing complex connectivity patterns that involve integrative processing or dynamic modulation.

**Physiological data**. We next investigate dependencies in a dataset of breathing and cardiac activity. While these two processes are known to be biologically linked (Adrian et al., 1932), the correlation between respiration rate and electrocardiogram (ECG) signals is approximately zero (Fig. 5a). We applied the proposed method to a dataset of 60 pairs of temporally synchronized ECG and respiration rate signals, collected at the Children's Hospital of Philadelphia using Zephyr BioModule sensors. The duration of the tasks used for data collection ranged from 4 to 7 minutes. The data were split into four subject-independent cross-validation folds. The segment size $w$ was equivalent to 5 seconds. The average concurrence coefficient on the test folds was 0.50 (p<0.001), indicating that the concurrence approach successfully detects the relationship between respiration rate and ECG signals. The PSCS can generally distinguish between compared signals that are temporally aligned or not (Fig. 5b-d), validating that concurrence can identify relationships (or lack thereof) that are difficult to determine visually. Fig. 5e plots the PSCSs from temporally aligned segments vs. the root mean square of successive differences (RMSSD) derived from the ECG signal of each interaction. That the PSCS is

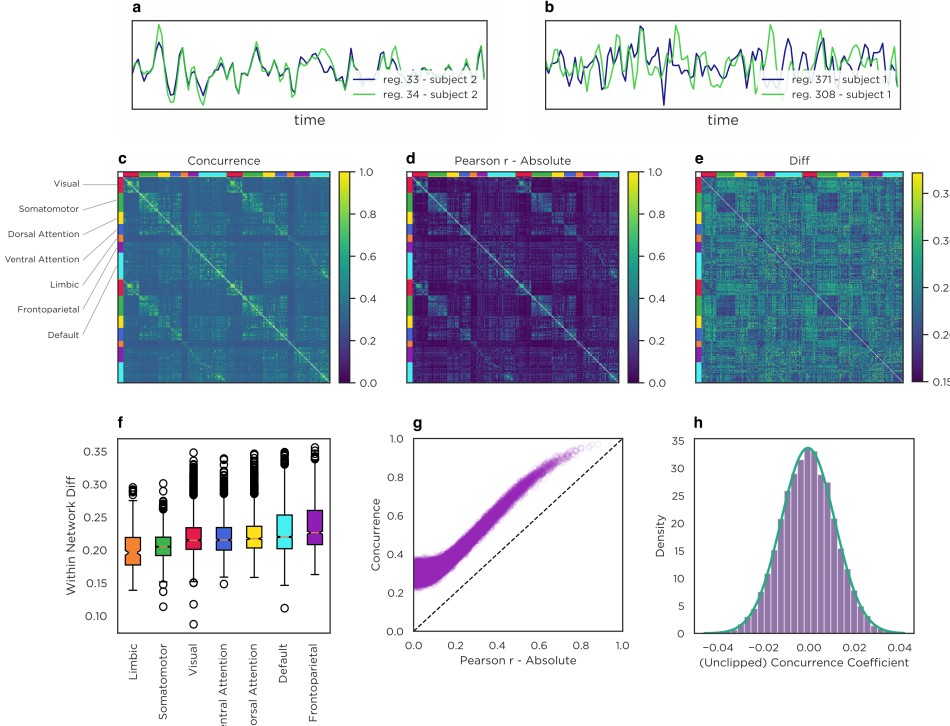

Figure 4: (a) Correlated fMRI signals from two brain regions. (b) Signals from regions that are dependent (concurrence coefficient: 0.25) but uncorrelated (Pearson's r: 0.02). (c) Connectivity matrix computed with the concurrence coefficient. (d) Connectivity matrix computed with (absolute) Pearson's $r$ values. (e) The difference between the concurrence- and correlation-based connectivity matrices. (f) The distributions of the difference between the concurrence- and correlation-based connectivity matrices, shown separately for the seven brain networks. (g) Comparison of the Pearson's r vs. concurrence coefficients computed from all the brain region pairs. (h) The (unclipped) concurrence coefficient between 10,000 pairs of brain regions of mismatched participants.

generally larger when the RMSSD is low may suggest that the trained algorithm predicts a stronger relationship between ECG and respiration rate when the latter is increased.

**Behavioral Data**. Finally, we apply concurrence to the analysis of facial behavior in a dyadic conversation. The behaviors of two conversation partners are expected to be dependent, due to well-known phenomena like nonconscious mimicry (Lakin et al., 2003) or (nonverbal) backchanneling (Shelley & Gonzalez, 2013). However, quantifying such dependencies has proven challenging, as behavior is captured with multi-dimensional signals (Supp. Fig. F.1), and any subset of signals from one conversation partner may depend on the signals of the other partner through an unknown relationship. We conduct experiments on a dataset of 199 participants (aged 5 to 40 years) engaged in a 3-5-minute semi-structured face-to-face conversation (Sariyanidi et al., 2023). We quantify social behaviors (i.e., facial expressions and head movements) in each conversation partner with 82-dimensional signals (79 for facial expressions and 3 for head movements) (Sariyanidi et al., 2024). The concurrence coefficient for $w = 4$ seconds is 0.49 (p<0.001), indicating that the behavior signals of the conversation partners are dependent. Moreover, the PSCS allows us to investigate differences within different subsamples. For example, PSCS increases with age (Spearman's r = 0.61, p < 0.001; Supp. Fig. F.1c), indicating that younger school-age children tend to have less behavioral coordination than older children. Additional analyses on a subsample of 12-18 year-olds (N=42) with and without an autism diagnosis (matched on age and sex) indicate that autistic adolescents have reduced coordination with conversation partners relative to neurotypical adolescents (Cohen's D: 0.8; p=0.003; Supp. Fig. F.1c). These results show that concurrence exposes clinically relevant differences in spontaneous behavioral coordination, without any a priori information about the structure of the coordination.

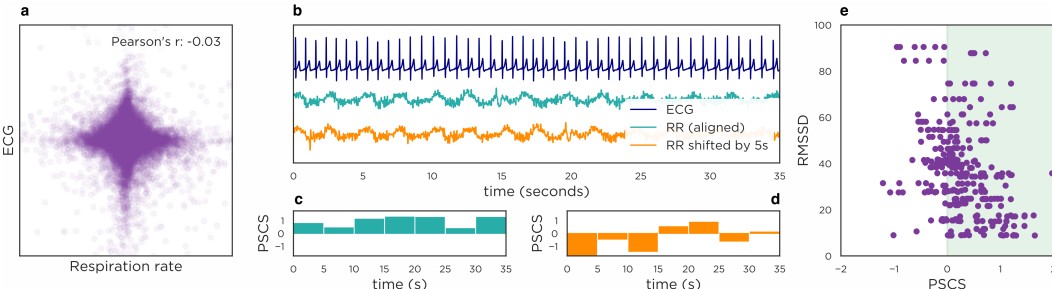

Figure 5: (a) Scatter plot and correlation between the respiration rate (RR) and ECG signal. (b) A sample ECG signal plotted against the synchronized (i.e., time-aligned) RR signal and the temporally misaligned RR. (c) The per-segment concurrence scores (PSCSs) between the temporally aligned ECG and RR are positive, which indicate that the PSCS correctly predicts that the segments are temporally aligned. (d) The PSCSs between the temporally misaligned ECG and RR are generally negative. (e) The PSCS for (temporally aligned) ECG and RR signals against the RMSSD, computed on a dataset of 30 participants for multiple segments per participant.

## 5 LIMITATIONS AND FUTURE WORK

We showed that the concurrence coefficient is proportional to the degree of dependence between the compared signals. However, one must exercise caution when making comparative judgments to ascertain whether a pair of processes are more strongly related than another pair, since the concurrence coefficient can be made larger by simply increasing the segment size $w$ (Section 2.2; Fig. 2c,d). As such, $w$ must be chosen in a way that the comparisons are commensurate. Nevertheless, the choice of $w$ is of little concern if one aims to ascertain whether two processes are dependent or not.

Our theoretical analyses relied on simplifying assumptions for analytical tractability. However, the proposed framework can be used beyond these assumptions. For example, the physiological signals in our experiments are highly periodic, which violates the (Bernoulli) assumption that consecutive dependence events are independent of one another (Bertsekas & Tsitsiklis, 2008). A critical future direction is to precisely delineate the dependencies that can be exposed through concurrence by relaxing or eliminating these assumptions. Finally, we implemented concurrence via CNNs as they are easy to train. However, in the presence of long-range relationships, CNNs may require parameter modification (Appendix A) or fail, whereas architectures such as transformers are highly capable in this scenario. The research by the machine learning community can lead to streamlined and efficient training procedures for more modern architectures, enabling their usage with concurrence.

## 6 CONCLUSION

We introduced a new approach for measuring statistical dependence in time series, namely, concurrence. We showed that constructing a binary classifier that simply distinguishes between concurrent and non-concurrent segments of the compared time series leads to a theoretically supported and practically potent framework. Concurrence can become a standard way of quantifying statistical dependence between time series, as it readily detects a wide range of linear or non-linear dependencies with an off-the-shelf implementation, even from modestly sized samples and noisy data, without requiring empirical (hyper)parameter tuning; showed no propensity to false discoveries (Type I errors), and works with single- or multi-dimensional signals. Future research can further enhance this framework by theoretically establishing the most general conditions under which concurrence exposes dependence, while integrating new architectures and well-established training recipes developed by the machine learning community can ensure that its theoretical potential can be fully actualized.

ACKNOWLEDGMENTS

Large language models (LLMs) have been used to aid the literature review of this paper.

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
