# OpenReview forum: "Concurrence: A dependence criterion for time series, applied to biological data"
_ICLR.cc/2026/Conference — Submitted to ICLR 2026_

### Official Review · Reviewer_4reK · 2025-10-26

**Soundness:** 2
**Presentation:** 2
**Contribution:** 1
**Rating:** 2
**Confidence:** 4

**Summary:**

The work introduces a metric termed “concurrence” to quantify statistical dependence between time series. The metric is evaluated as a CNN classifier using CL. It is applied to three data sets: fMRI, breathing/heart rate, facial expressions/head movements. The metric is compared on synthetic data to eight other metrics commonly used for statistical discrimination. It is claimed “Concurrence can become a standard way of quantifying statistical dependence between time series”.

**Strengths:**

Recognition of the distinction between non-linear dependence and linear independence.

The method and implementation are claimed to be “easy to use” and “open source”.

The method claims to operate “without requiring empirical (hyper)parameter tuning”.

It is claimed to “expose relationships across a wide spectrum of signals”.

The method is demonstrated on three real world data sets.

**Weaknesses:**

Although statistical dependence is useful, it is limited to probabilistic descriptions rather than state-dependent, mechanistic relations. Since non-linear dependence is fundamentally state-dependence, statistical dependence is arguably less useful when untangling complex system dynamics. Statistics describe mean fields, while state transitions define operational, mechanistic relations. The article makes no mention of this.

While statistical descriptions are useful for characterizing populations and relationships of distributions, identifying directional, state-dependent relationships providing mechanistic insight and explainability are current goals in ML. The article does not make a convincing argument that applying CCN/CL to identify statistical dependence, with significant computational overhead  (linear scaling) while disallowing time-lag intra-dependencies, significantly advances tools of machine learning or scientific discovery.

The method identifies a dependence coefficient, but does not untangle directionality of the dependence.

Real world complex systems and networks routinely encapsulate time-lagged causal interactions. Concurrence does not allow such dependencies.

The synthetic data sets are simplistic presuming a simple mathematical functional dependence (applying a kernel transformation) provides a meaningful and robust set of “dependent” time series.  This is too simplistic to claim general robustness, a wider survey of truly state-dependent systems is needed.

Applying the method to three real world data sets falls significantly short of supporting the claim of robustness to “a wide spectrum of signals”.

As the method is based on CNN and probabilistic renderings, there is no direct explainability or state-dependent mechanistic insight.

No source code or library is provided to independently evaluate claims.

**Questions:**

1) Line 098
“If both xt,w and yt,w contain responses to a common event, then they must be statistically dependent.”

This is likely an overstatement. Especially in the realm of non-linear, episodic dynamics.  While it is clear there should be state-dependence, presuming a statistical dependence is not. It is easy to imagine situations where x,y contain “responses”, but those responses are not amenable to statistical characterization.


2) Line 103
“This idea is not only intuitive, but also theoretically plausible.”

This suggests coincidence is a valid dependence metric, and presumes statistical dependence can be identified in non-linear systems. We know statistical dependence presumptions are often ill-posed in non-linear systems.


3) 118-119
“The two segments are deemed concurrent if the PSCS is positive, and not concurrent if it is nonpositive.”

“nonpositive”?


4) 118 119
“Kx and Ky are the dimensions of x and y”  “The PSCS is computed by first transforming the input segments via separate learned functions f and g into segments of dimensions Kf and Kg”

Why are dimensions not characterized? How are they selected?


5) 151 - 158
The basis is formulated on discrete time random sequences, then imposing dependence “processes hx[t] and hy[t] are modeled as the product of a common process h[t] with two separate and independent processes α[t] and β[t] that take binary values (0 or 1)” … “The common process h[t] ensures that x[t] and y[t] are dependent”

I find this to be a contrived, created dependence, not a true state-dependence manifest in real world non-linear systems.


6) Equation (11) is peculiar.

Why are all these seemingly equivalent statements repeated?


7) 238
“complexity of our implementation is O (N w(Kx + Ky ))”

Should it be (N^2 – N)/2 to account for computation of all pairwise interactions and since there is no directional distinction? O(N..) is a linear scaling where training a CNN is considered a single step? It seems peculiar to assign training a CNN as a computational complexity of 1 for each time series pair. I did not find quantified mention of dimensions K.


8) The method is computationally intensive.

For a small system of N = 400, T = 1000 it is stated to require 40 seconds on 10,496 CUDA cores @ 1.4 GHz. What is the corresponding runtime for the other metrics? Contemporary problems may have N = 1E6, T = 1E4, is the computational overhead tractable at this scale, especially in relation to other metrics?


9) Figure 5. d) The PSCSs between the temporally misaligned ECG and RR are generally
negative.

This is not supported by the figure.


10) Figure 5. e) The PSCS for (temporally aligned) ECG and RR signals against the RMSSD, computed on a dataset of 30 participants for multiple segments per participant.

It is stated: “That the PSCS is generally larger when the RMSSD is low may suggest that the trained algorithm predicts a stronger relationship between ECG and respiration rate when the latter is increased.”

This is not evident in the figure.

---

### Official Review · Reviewer_RVoa · 2025-10-30

**Soundness:** 2
**Presentation:** 3
**Contribution:** 1
**Rating:** 2
**Confidence:** 4

**Summary:**

The authors propose a measure for statistical dependency for time series, building on temporal concurrence. The measure is similar to the contrastive objective used e.g. in independent component analysis, learning a classifier to separate between temporally aligned segments and misaligned segments and measuring the dependency with (a monotonic transformation) of the classifier accuracy. The measure is analysed from the perspective of detecting deviation from complete independence, and empirical validation focusing on this is provided.

**Strengths:**

Measuring non-linear statistical dependency is not trivial, especially for structured data such as time series. The problem setup is hence worth investigating and relevant for the community.

The proposed method is easy to understand and implement, requiring only training of a simple classifier. It is intuitive how classification accuracy better than chance level is a sign of some kind of dependency.

**Weaknesses:**

The paper has several weaknesses that render it unpublishable in top-tier venues.

The most critical one is that it fails to provide a real measure of dependency. The reasoning is perfectly valid for separating between independence and some degree of dependence, but the paper omits a proper justification on why exactly 2*accuracy-1 would be a good measure of dependency in a general case. The authors seem to be well aware of this limitation, explicitly saying that the coefficient is *"proportional to the degree of dependence"*. Unfortunately it is a major limitation; quantifying the degree of dependence is always the harder part without that we should not call any approach a dependency criterion or metric. If the method can only detect presence or absence of dependency, it should be characterised as a statistical test of independence. In other words, the paper falls short of what is promised in the title and the abstract that talk about measuring the dependency. The paper might work better if written specifically to describe a statistical test of independence.

This issue influences the whole paper. Both theoretical and empirical evidence focuses on verifying the method can identify also weak dependences, but the arguably more interesting question of quantifying the dependency is left open. For example, Table 1 clearly shows the method is able to better detect the dependencies, but the experiment is left one-sided: There is no discussion of possible false positives and if I understood the experiment right a hypothetical method always indicating dependency would look perfect in this evaluation.

The scientific novelty is also limited. Like the authors admit, the idea of learning a classifier to distinguish between temporally aligned and misaligned segments is not new and the way it is done in this paper is straightforward. Moreover, the previous uses are effectively in the same tasks: the whole point of contrastive methods for ICA is to accurately characterise independence. Note that in their use case the lack of quantification of dependence is not an issue as it is used purely as a learning objective, and hence the previous works do not share the main limitation of this paper.

**Questions:**

How does the performance of the method depend on the architecture of f and g? Some sort of ablation study on that would be useful for understanding when it starts to fail.

---

### Official Review · Reviewer_CC6n · 2025-10-30

**Soundness:** 2
**Presentation:** 2
**Contribution:** 2
**Rating:** 2
**Confidence:** 4

**Summary:**

The paper proposes a particular form of noise contrastive estimation (NCE) for time series dependence testing. Mathematically, I interpret the NCE form as
$\mathbb{E}[ \log h((x_{t,w},y_{t,w};\theta) )] + \mathbb{E}[ \log(1- h((x_{t,w},y_{t',w};\theta) ) )]$  where $h((\cdot,\cdot) ) = \frac{1}{1+e^{-\langle A , \mathrm{Cov}(f(\cdot),g(\cdot) )\rangle}}$ where $A_{ij}=\alpha_{ij}, f, g$ are learnable weights and two embedding functions $f:\mathbb{R}^{K_x\times w}\rightarrow \mathbb{R}^{f\times w'}$ and  $g:\mathbb{R}^{K_y\times w}\rightarrow \mathbb{R}^{g\times w'}$, respectively, and $\mathrm{Cov}$ computes the empirical covariance based on a embedding duration  of size $w'$. This can be seen as learning a non-linear embedding  and that maximizes a sort of canonical correlation analysis score to distinguish paired and unpaired time segments.  Results on synthetic data shows the statistical power and results on brain fMRI show potential for this generalized correlation.

**Strengths:**

Generalized correlation analysis for time series is a problem of significance. The interpretable score that is the logit is interesting. The statistical power on synthetic data looks promising. The brain imaging example is interesting.

**Weaknesses:**

In Section 2.1, the discussion of the computation and use PSCS for specific pairs is not clear. As in my summary above I assume the empirical covariance is computed per window, and the binary cross entropy is applied to what $s$ as logit. There should be an explicit use of $s$ in the definition of the measure.

It is not clear if variable $w$ (with resulting $w'$) be used? If so that is a benefit for comparing possibly unequal sized segments.  I would assume the 1D convolution and pooling is used to reduce from $w$ to $w'$.


The theoretical section 2.2 seems limited by assumptions and clarity, and contrived. The assumptions at line 146 and 147 should be made clearer. It's not clear why the Bernoulli process model is representative of the diversity of multivariate stochastic processes.

The paper misses the literature on classifier-based two-sample tests [a], which provide divergence measures between the joint and product of the marginals as a measure of dependence.  For instance in Lopez-Paz and Oquab independence testing is performed for sinusoidal signals. More recent work looks at classifier based independence testing in an online sequential decision making setting, and optimization of kernels via efficient representations like random Fourier features [c].  It is discouraging that LLMs were used for literature review, given the availability of other ways to read and explore relevant literature.

[a] Lopez-Paz, David, and Maxime Oquab. "Revisiting Classifier Two-Sample Tests." In International Conference on Learning Representations. 2017.
[b] Podkopaev, Aleksandr, and Aaditya Ramdas. "Sequential predictive two-sample and independence testing." Advances in neural information processing systems 36 (2023): 53275-53307.
[c] Ren, Yixin, Yewei Xia, Hao Zhang, Jihong Guan, and Shuigeng Zhou. "Efficiently learning significant fourier feature pairs for statistical independence testing." Advances in Neural Information Processing Systems 37 (2024): 99800-99835.

Benchmarks should include [a] and [c] and ablations based on directly optimizing a discriminator to distinguish paired and unpaired windows but with similar architecture (without the $\mathrm{Cov}$). Also an ablation that uses the loss in CLIP, (embedding each window to fixed length vector and then using cosine similarity rather than maintain temporal dimension and compute the scored from weighted combination of the covariance entries).

**Questions:**

Why is the Bernoulli process model representative of the diversity of multivariate stochastic processes?

Line 168, Why would the convolution of kernels make the dependence non-linear? Instead it is the modulating by $alpha[t]$ and $\beta[t]$.

Given the high degree of dependence in Figure 4(g) between linear correlation and Concurrence, it is not clear what is novel associations/dependence revealed by this method? The values simply are elevated.

Minor:

Concurrence lines 39 Odd spacing around the em-dashes.

---

### Meta-Review · Area_Chair_f8pJ · 2025-12-26

**Summary:**

The submission introduces Concurrence, a dependence criterion for time series that operationalizes dependence via the ability of a learned classifier to distinguish temporally aligned from misaligned segment pairs, and reports promising detection results on synthetic benchmarks and several biological datasets (fMRI, physiology, and behavior). Reviewers generally agreed the high-level intuition is reasonable and the empirical demonstrations are potentially interesting, but they also converged on the view that the paper, as written, does not meet the bar for a top-tier venue. In particular, the work is judged to be closer to an independence test than a principled, interpretable dependence measure, and the novelty and evaluation framing are not sufficiently convincing to support the breadth of claims.

**Reviewer Concerns:**

A central, repeated concern is conceptual: the paper does not provide an adequate justification for why the proposed normalization of classifier accuracy (i.e., a bounded “concurrence coefficient”) should meaningfully quantify the degree of dependence in general, as opposed to merely detecting deviations from independence. Relatedly, the dependence of the score on design choices such as segment length raises comparability issues and undermines interpretability, and the method does not address directionality or time-lagged dependencies that are common in real-world dynamical systems. Reviewers also questioned the theoretical section for relying on strong, arguably contrived assumptions (e.g., Bernoulli/stationarity and no lag) and for not clearly establishing generality. On positioning and novelty, reviewers noted substantial overlap with existing contrastive and classifier-based two-sample/independence testing literature and requested stronger engagement with that prior work, as well as clearer ablations against simpler discriminators and alternative contrastive objectives. Finally, the experimental validation was considered one-sided in places (emphasizing power on synthetic data without a correspondingly clear analysis of false positives), with additional concerns about unclear methodological description (e.g., PSCS definition/usage), questionable interpretation of some plots, computational overhead and scalability, and the lack of released code despite “open-source” framing.

**Reviewer Scores:**

Given that the core objections concern the fundamental framing (measure vs. test), theoretical generality, and limited incremental novelty, it is unlikely that further discussion would have materially increased scores beyond minor presentation-related adjustments; the expected overall recommendation would remain rejection.

---

### Decision · Program_Chairs · 2026-01-26

Reject